# Older Adults with Hypertension: Prevalence of Falls and Their Associated Factors

**DOI:** 10.3390/ijerph18168257

**Published:** 2021-08-04

**Authors:** Atirah Az-Zahra Abu Bakar, Azidah Abdul Kadir, Nur Suhaila Idris, Siti Nurbaya Mohd Nawi

**Affiliations:** 1Department of Family Medicine, School of Medical Sciences, Universiti Sains Malaysia, Kubang Kerian 16150, Malaysia; atirahazzahra@student.usm.my (A.A.-Z.A.B.); nursuhaila@usm.my (N.S.I.); 2Department of Internal Medicine, School of Medical Sciences, Universiti Sains Malaysia, Kubang Kerian 16150, Malaysia; nurbayanawi@usm.my

**Keywords:** falls, hypertension, older adults, older people

## Abstract

Falls are prominent health issues among older adults. Among hypertensive older adults, falls may have a detrimental effect on their health and wellbeing. The purpose of this study is to determine the prevalence of falls among hypertensive older adults and to identify the associated factors that contribute to their falls. This was a cross-sectional study conducted among two hundred and sixty-nine hypertensive older adults who were selected via systematic random sampling in two primary health clinics in Kuala Terengganu, Malaysia. Data on their socio-demographic details, their history of falls, medication history and clinical characteristics were collected. Balance and gait were assessed using the Performance Oriented Mobility Assessment (POMA). It was found that 32.2% of participants reported falls within a year. Polypharmacy (adjusted OR 2.513, 95% CI 1.339, 4.718) and diuretics (adjusted OR 2.803, 95% CI 1.418, 5.544) were associated with an increased risk of falls. Meanwhile, a higher POMA score (adjusted OR 0.940, 95% CI 0.886, 0.996) and the number of antihypertensives (adjusted OR 0.473, 95% CI 0.319, 0.700) were associated with a low incidence of falling among hypertensive older adults. Falls are common among hypertensive older adults. Older adults who are taking diuretics and have a polypharmacy treatment plan have a higher incidence of falls. However, older adults taking a higher number of anti-hypertensive medications specifically were not associated with an increased prevalence of falls.

## 1. Introduction

The older adult population is projected to more than double, reaching more than 1.5 billion people by 2050. One in six people worldwide will be aged 65 years or over. As the world’s population of older adults is expanding at an alarming rate, falls have become one of the major health concerns. Previous prevalence studies on falls among older adults have shown variable results. In the United States, 12% of older adults recorded falls [1]. Meanwhile in South Africa, 26.4% of older adults from suburban communities experienced falls [2]. Among Asian countries, Indonesia had a prevalence of 12.8% for falls among older adults [3] whereas a study in India revealed 24.98% of their community-dwelling older adults experienced falls [4]. However, there was a low prevalence of falls among community-dwelling older adults in Malaysia, which was 4.07% [5]. This figure differs significantly from the prevalence of falls among older adults with comorbidities, as seen in the research of diabetic older adults where the prevalence of falls was 18.8% [6]. Falls may cause physical injuries such as osteoporotic fractures and head injuries leading to hospitalization and mortality. This contributes to increased healthcare costs [7,8]. Even in the absence of physical injuries, falls may cause long-term psychological consequences including depression and a fear of falling in addition to anxiety, a loss of self-confidence and activity avoidance. These psychological consequences subsequently lead to the restriction of their daily and social activities [7,9,10].

Hypertension is one of the worrisome non-communicable diseases among older adults [11]. According to the World Health Organization’s Study on Global Ageing and Adult Health, the prevalence of hypertension in the population aged 50 years and above in middle- and lower-income countries is 52.9% [12]. In Malaysia, the prevalence of hypertension among older adults over 60 years old is also showing an increasing trend, rising from 14.3% in 2006 to 19% in 2015 [13].

Numerous concerns have been reported regarding hypertension and falls among hypertensive older adults. Older adults who are on treatment for hypertension are known to be at risk of falls and fall-related injuries [14,15]. Orthostatic hypotension (OH) is a common clinical issue that is linked to hypertension [16]. OH assessed in a middle-aged, community-based population was also implicated with the risk of hospital- or healthcare-related falls. Postural reductions in either the systolic or diastolic blood pressure or both showed an inverse relationship with falls [17]. Older adults with an elevated systolic blood pressure have greater postural changes in blood pressure than those with a controlled systolic blood pressure [18]. Apart from this, hypertensive older adults with a moderate to high exposure to multiple antihypertensive drugs are likely to experience more significant reductions in their systolic blood pressure. The categorical analysis also suggested an increased fall risk in women with a diastolic BP lower than 70 mmHg [19] whereas other research has reported an increased risk of falls in male participants with either low systolic or low diastolic blood pressure [20]. The use of diuretics was found to be associated with a higher incidence of falls due to the side effect of postural hypotension [21,22]. Beta blockers were significantly associated with a decreased fall risk [23].

Although hypertensive older adults are more prone to experiencing falls, there are still inadequate data regarding falls. Despite earlier studies among older adults with comorbidity showing a considerable prevalence of falls, there are still no local data on falls and hypertensive older adults in Malaysia. Their associated factors can be used to construct a structured plan for a fall intervention and prevention methodology in this population. The objective of this study is to determine the prevalence of falls among hypertensive older adults and to identify the factors that are associated with those falls.

## 2. Materials and Methods

### 2.1. Study Design and Population

This study was a cross-sectional study conducted among 269 participants from December 2019 to February 2020 in two primary health clinics in Kuala Terengganu on the east coast of Peninsular Malaysia. The calculation for the sample size of all the objectives of this study was conducted and the biggest sample size calculated was used for this study. The sample size selected for this study was elicited from the sample size to determine the prevalence of falls among hypertensive older adults and was calculated using the single proportion formula. By taking the proportion of falls as 0.2 [24], using a precision rate of 0.05 and providing a leeway of 10% for non-responsive participants, the calculated sample size was 269. The participants were chosen using systematic random sampling in a ratio of 1:5 based on attendance lists in Kuala Terengganu primary health clinics until the calculated sample size was reached.

The research instruments consisted of a case report that divided the area into six sections. The Section 1 consisted of The Elderly Cognitive Assessment Questionnaire (ECAQ). The Section 2 of the case report was a socio-demographic survey that included the participant’s socio-demographic data such as age, gender and living arrangements. The clinical data questionnaire in the Section 3 inquired about the type of chronic disease they had, their medication list and polypharmacy practice details. This was in addition to the number of falls they had in the past year and the circumstances and complications related to the fall episode if present. The Section 4 of the form consisted of the results of the physical examination, which included body mass index (BMI), blood pressure (sitting and standing) and visual acuity. The Section 5 was the functional assessment utilizing the Barthel index questionnaire. The Section 6 consisted of a balance and gait assessment. The use of assisted mobility devices was also recorded. 

A fall is defined as an unanticipated circumstance in which the participant comes to rest on any surface such as the ground, floor or a lower level without a loss of consciousness [25]. Older adults are defined as having a chronological age of 65 years or older [26]. The inclusion criteria were: patients aged 65 years old or older, those who had been diagnosed with hypertension for at least one year and those who were on treatment for hypertension for at least two weeks. Non-ambulatory older adults who were unable to walk with or without a walking aid and those with a cognitive impairment whose caregivers were not present during the clinical assessment were excluded from the study. Orthostatic hypotension (OH) is defined as a decrease in the systolic blood pressure (BP) of at least 20 mmHg or the diastolic BP of at least 10 mmHg with a transition from a sitting to a standing position one minute apart [27,28]. The participants were classified as having visual impairments if their visual acuity result was 6/18 or worse [29]. Polypharmacy was defined as the use of ≥5 medications [30]. The Barthel index questionnaire was used to measure the participant’s level of independence while performing basic activities of daily living such as eating, washing, getting around and sphincter control. The scale is from 0 to 100 points. The Performance Oriented Mobility Assessment (POMA) comprises nine balance tasks and seven items used to assess gait characteristics [28]. The balance components are sitting balance, standing, attempts to stand, immediate standing balance, standing balance, nudged, eyes closed, turning 360 degrees and sitting down. This comes to a total of 16 points. The gait elements consisted of a commencement of gait, step length, step symmetry, step continuity, path, trunk and walking attitude for a maximum of 12 points. Each subscale was measured as either abnormal = 0 or normal = 1 or, in several cases, adaptive = 1 and normal = 2.

### 2.2. Data Collection

The participants were approached and the study details were explained thoroughly by the researcher. Written consent was obtained from either the selected participants or their caregivers. Face-to-face interviews with participants were conducted in the primary health clinic by the researcher to complete the case report form. Blood pressure measurements at both a sitting and standing position within one minute apart, a height and weight measurement, a functional assessment using the Barthel Index, a visual acuity examination using the Snellen chart and a Performance Oriented Mobility Assessment were performed. The use of assistive devices was recorded. The researcher reviewed the participants’ medical records to obtain their medical information including their comorbidities and medication list.

### 2.3. Statistical Analysis

The analysis of the collected data was done using SPSS software (SPSS Inc., version 24, IBM Corp, New York, United States of America). The socio-demographic data, fall-related events and medical characteristics of the participants were analyzed using the mean, standard deviation, frequency and percentages. A simple logistic regression was used to analyze the categorical variables. All variables with a *p*-value less than 0.25 were included in the multiple logistic regression analysis. Backward and forward stepwise procedures were performed for all significant variables related to falls in order to look for possible two-way interactions. The final model to determine the independent effect on falls was performed by adjusting for confounders, specifically age and gender. The associations were expressed as an odds ratio with a 95% confidence interval. *p*-values were considered to be statistically significant at *p* < 0.05. The goodness of fit model was analyzed using Hosmer and Lemeshow, the classification table and the receiver operating characteristic curve.

## 3. Results

### 3.1. Characteristics

A total of 288 subjects who fulfilled the inclusion criteria were approached to be involved in this study. Only 269 participants consented to participate. The response rate was 93.5%. Table 1 shows the demographic and clinical characteristics of the participants. The mean age of the participants was 71.00 (±5.24). The majority of fallers were older adults from the age group 65–69 years old. Furthermore, most of the fallers were female. The ECAQ and Barthel index scores were nearly comparable for both fallers and non-fallers. Fallers scored less in the Performance Oriented Mobility Assessment compared to non-fallers. The number of antihypertensives used by fallers and non-fallers did not differ significantly. Two-thirds of those who fell used a calcium channel blocker as one of their hypertension medications.

### 3.2. Prevalence and Characteristics

The prevalence of falls among hypertensive older adults was found to be 32.2% (n = 90). Among the fallers, 60% were female and 40% were male. Most of the falls (48.9%) took place inside the house whereas 13.3% (n = 12) of fallers experienced falls both inside and outside the house. A total of 77.8% of falls occurred during the daytime with 33% of the falls causing mild soft tissue injuries. 

### 3.3. Simple Logistic Regression

Table 2 shows the simple logistic regression analysis for all falls factors that were seen among hypertensive older adults in this study. There was no correlation between fall prevalence and gender, ECAQ score, visual impairment, heart disease, stroke, hyperlipidemia and arthritis in the simple logistic regression analysis. There was no significant number of hypertensive older adults who fell when taking an angiotensin-converting enzyme (ACE) inhibitor, angiotensin receptor blockers (ARB) or an alpha blocker as their hypertension medicine. Blood pressure parameters and orthostatic hypotension were also found to be unrelated to falls. The data with a *p*-value < 0.25 were found to be correlated with the hypertensive older adult aged 70–74 years old who were living alone, their Barthel index score, polypharmacy, number of medications, number of antihypertensives, POMA score, diabetes, use of calcium channel blockers, diuretics and beta blockers. The data were subsequently analyzed using a multiple logistic regression. 

### 3.4. Multiple Logistic Regression Analysis of the Significant Variables and Their Relation to Falls among Hypertensive Older Adults

Table 3 shows the multiple logistic regression analysis of the significant variables and their relation to the falls among hypertensive older adults. Polypharmacy (OR 2.513, 95% CI 1.339, 4.718) and diuretic use (OR 2.803, 95% CI 1.418, 5.544) were associated with a higher risk of falls. A higher Performance Oriented Mobility Assessment (POMA) score (OR 0.940, 95% CI 0.886, 0.996) and the number of antihypertensives (OR 0.473, 95% CI 0.319, 0.700) were both associated with a reduced risk of falls among hypertensive older adults.

### 3.5. Fitness of the Multiple Logistic Regression Models

To determine the fitness of the multiple logistic regression models, Hosmer and Lemeshow’s goodness of fit statistics revealed a *p*-value of 0.827. The sensitivity of the final model was 30% and the specificity was 90.5%. The overall percentage was good (70.3%). The area under the ROC curve was 0.327. 

## 4. Discussion

The prevalence of falls among hypertensive older adults in this research was 32.2%. It was substantially higher when compared to community-dwelling older people and specific populations of older adults with other comorbidities such as diabetes [6]. Furthermore, the prevalence of falls among hypertensive older people was equivalent to that of older adults with cancer [31]. This prevalence was within the range of the prevalence reported by previous studies related to falls among hypertensive older adults [24,32]. For comparison, Lipsitz et al. conducted an observational study of 598 hypertensive older adults in Boston, Massachusetts. They noted that the prevalence of falls among hypertensive older adults was 44.64% within the one-year follow-up. Similar to our study, the majority of the participants were female (62.2% vs. 59.5%). However, the high prevalence of falls in Lipsitz et al.’s study was due to the older mean age of the fallers compared to our study (78.5 vs. 71.2) [32]. Another study that was performed involving 170 hypertensive patients in the outpatient clinic of the University Hospital of Naples “Federico II” showed a lower prevalence of falls with 20% of the participants reporting falls within three months of the follow-up. The proportion of female participants was not comparable with our study despite the mean age of the participants being equivalent [24]. This demonstrated that older people with hypertension require special attention when it comes to falls.

Our study found that orthostatic hypotension was not a significant risk factor in falls among older adults with hypertension. The prevalence of OH was noted to be very low among the participants in the fallers group, which was 6.7% (*p* = 0.734). This contradicts numerous studies that have concluded that orthostatic hypotension is related to falls [17,33]. According to a previous opinion, the escalation of existing antihypertensive drugs or the addition of new antihypertensive medications are more likely to cause excessive postural hypotension [34]. In hypertensive patients, the occurrence of OH is postulated as a direct effect of the use of antihypertensives. Antihypertensive combinations including α blockers, centrally acting drugs, non-dihydropyridine calcium channel blockers and diuretics are associated with OH [35]. In the SPRINT (Systolic Blood Pressure Intervention Trial) study, which looked at the association between OH and cardiovascular and other adverse events, 8792 participants were randomly assigned to either intensive or standard hypertension treatments. Notably, this study found there was no association between OH and an increased risk of falls. However, the investigators excluded participants with a standing BP of less than 110 mmHg [36]. A review article by Zia et al. also concluded that the evidence for OH and falls induced by antihypertensives is weak [37]. Other studies also showed that having a controlled systolic blood pressure reduces the risk of orthostatic hypotension that subsequently causes a fall [38,39].

Polypharmacy is a known risk factor for falls found in this study. It was found that 73.3% (n = 66) of the fallers had polypharmacy. This finding was similar to another study by Kojima et al. [30]. Among the 172 participants (95.9% follow-up rate), polypharmacy was a significant predictor of falls according to the multivariate analysis (OR 4.50; CI 95% CI 1.66–12.2) [30]. As supported in the research conducted by Gnjidic et al., the use of ≥5 medications contributed to falls [40]. A longitudinal cohort study among 5312 English participants aged 60 and older by Dhalwani et al. found that almost one-third of the total population were using five or more drugs. It was also significantly associated with a 21% higher rate of falls over two years (adjusted IRR 1.21, 95% CI 1.11 to 1.31) [41]. The correlation between falls and polypharmacy is strengthened by the evidence showing that the use of ≥ 5 medications gives rise to poorer mobility and physical performance as well as cognitive impairments among older people [42,43].

Previous studies have shown there was an association between diuretic use and falls. This is consistent with our study where the participants who were on diuretics experienced a greater number of falls (OR 2.803; 95% CI 1.418–5.544). According to Gribbin et al. from their self-controlled case series analysis of 9862 individuals older than 60 years of age in the United Kingdom, thiazide diuretics were found to increase the risk of a first fall significantly (OR 1.25; 95% CI 1.15–1.36). No other antihypertensive showed a significant association with falls [21]. A systematic review and meta-analysis by De Vries et al., which analyzed 131 studies, showed that loop diuretics were significantly associated with an increased fall risk (OR 1.36; 95% CI 1.17, 1.57) [23]. A population-based self-controlled case series study by Butt et al. in Ontario involving 543,572 new users of antihypertensive drugs among community-dwelling older people identified that new antihypertensive users aged 66 and older had a 69% increased risk of having a fall (RR 1.94; 95% CI 1.75–2.1). The initiation of antihypertensives including thiazide diuretics increased the fall risk of older adult patients to the point of being significant during the first 14 days [44]

Gait and balance disorders have been consistently identified in multiple reviews as among the strongest risk factors for falls [25,45,46]. Alternatively, hypertension may increase the fall risk by affecting the person’s control of their gait and balance [47]. The univariate and multivariate analysis from our study showed there was a significant association between the Tinetti Performance Oriented Mobility Assessment (POMA) and fall events (OR 0.940; 95% CI 0.886, 0.996). Our study finding is comparable with a cross-sectional study by Shen et al. with a sample of 176 participants with a mean age of 76.7 ± 6.6. The study found that older adults with uncontrolled hypertension had an impaired standing balance [48]. The assessment of gait and balance is crucial when evaluating and stratifying the fall risk among hypertensive older adults.

From our study, we also found that an increasing number of antihypertensives (OR 0.473, *p* < 0.05) were associated with a lower fall prevalence among hypertensive older adults. This is a new finding indicating that a higher number of antihypertensives is a protective factor towards falls. According to Emanuelsson’s retrospective case-control study of 129 hypertensives, multi-ill patients receiving home care, having 3 or 4 different types of antihypertensive drug groups increases the risk of falling compared to having 0 to 2 different kinds of antihypertensive drug groups [49]. A previous study by Shimbo et al. demonstrated that antihypertensive drug commencement and intensification as well as the addition of new antihypertensives classes were linked to a short-term elevated risk of significant fall injuries among older people [50]. A nested case-control study among older Singapore residents (age ≥ 60 years old) with a low socio-economic status (n = 210) reported that those on ≥ 2 antihypertensive medication types had a higher risk of having an injurious fall than those who were not on any antihypertensive medication (OR 5.45; CI: 1.49–19.93; *p* = 0.01) [51]. However, in a population-based cohort study by Bromfield et al. with a sample of 14,961 participants aged 65 years and older, the number of antihypertensive medications being taken was found not to be associated with an increased risk of serious fall injuries even after a multivariable adjustment [52]. Zia et al. conducted a case-control study with a sample of 352 participants and found there to be no statistically significant association between the number of antihypertensives and falls after a multivariate adjustment for age and the number of comorbidities [OR 1.6; CI 0.95–2.6] [22]. We note that taking a greater number of antihypertensives contributes to polypharmacy practices, which is also a predictor of falls. However, this study demonstrated that although the number of antihypertensives contributes to polypharmacy, it did not associate with falls. Further research may be needed to analyze the types of drugs that contribute to falls among those older people with polypharmacy instead of considering the number of medications.

## 5. Limitations of the Study

Our study has a few limitations. Firstly, we did not observe the compliance towards antihypertensives or other medication during the study period. We may not have been able to detect orthostatic hypotension if the participants did not take their medication as prescribed. The doses of the antihypertensive medications were not studied further in this research. The technique of checking the postural BP (supine vs. sitting to standing) or the timing of checking standing BP, e.g., after 1 min vs. 3 min, may affect the detection rate of orthostatic hypotension. It is worth noting that the ROC outcome was underwhelming (0.327). This suggests that there are other factors such environmental and psychological influences that may contribute to falls in older people with hypertension that were not examined in this study.

## 6. Conclusions

There is high prevalence of falls among older adults with hypertension in primary care centers. Falls are associated with multiple factors such as impairments in gait and balance, polypharmacy and diuretic use. Medical practitioners may need to conduct a thorough assessment to evaluate the need of prescribing diuretics as well as other types of medication because an inappropriate pharmacological treatment might contribute to falling among hypertensive older adults. The simplified version of the gait and balance test should be performed on hypertensive older adults who have risk factors such as polypharmacy and diuretic use. The addition of a new class of antihypertensives should be made if it is clinically indicated without excessive concern about the risk of falling as increasing numbers of antihypertensive drugs are not connected with an increased incidence of falls. A fall risk assessment and fall interventions should be implemented in the hypertensive older adult population because they are at an increased risk of falls. A targeted fall prevention program can be executed in primary care centers as these centers cater for much of the medical care of this population. Future research on interventions related to the identified risk factors may be performed to investigate if further falls can be prevented among older adults with hypertension. There is a need to design and commence fall screening and an awareness program to modify the risk factors that will improve the quality of life of hypertensive older adults. 

## Figures and Tables

**Table 1 ijerph-18-08257-t001:** Demographic and medical characteristics of the study participants.

Risk Factors	Fallers (n = 90)	Non-Fallers (n = 179)
Age group, n (%)		
65–69	37 (41.1)	90 (50.3)
70–74	35 (38.9)	42 (23.5)
>75	18 (20.0)	47 (26.3)
Gender, n (%)		
Male	36 (40.0)	73 (40.8)
Female	54 (60)	106 (59.2)
Living arrangement, n (%)		
With others	84 (93.3)	173 (96.6)
Alone	6 (6.7)	6 (3.4)
Medical characteristics		
ECAQ score, mean ± SD	8.59 ± 1.14	8.58 ± 1.08
Barthel index, mean ± SD	95.00 ± 10.34	97.03 ± 6.41
Visual impairment, n (%)	48 (53.3)	88 (49.2)
Polypharmacy, n (%)	66 (73.3)	109 (60.9)
Number of medications used by participants, mean ± SD	5.51 ± 1.86	5.16 ± 2.00
Number of antihypertensives used by participants, mean ± SD	2.12 ± 0.83	2.35 ± 1.03
POMA score, mean ± SD	23.77 ± 5.98	25.35 ± 3.42
Medical comorbidities, n (%)		
Diabetes mellitus	62 (68.9)	106 (59.2)
Heart disease	12 (13.3)	24 (13.4)
Stroke	5 (5.6)	5 (2.8)
Hyperlipidemia	87 (96.7)	170 (95)
Arthritis	48 (53.3)	89 (49.7)
Types of antihypertensive, n (%)		
ACE inhibitor	49 (54.4)	99 (55.3)
Calcium channel blocker	60 (66.7)	139 (77.7)
Diuretics	46 (51.1)	76 (42.5)
Angiotensin receptor blocker	15 (16.7)	36 (20.1)
Alpha blocker	3 (3.3)	10 (5.6)
Beta blocker	16 (17.8)	49 (27.4)
Blood pressure (mmHg), mean ± SD		
Sitting		
Systolic blood pressure	153.31 ± 18.30	156.00 ± 18.06
Diastolic blood pressure	83.95 ± 9.15	84.20 ± 11.68
Standing		
Systolic blood pressure	156.03 ± 18.45	158.34 ± 19.66
Diastolic blood pressure	89.93 ± 9.82	88.99 ± 11.05
Orthostatic hypotension, n (%)	6 (6.7)	14 (7.8)

**Table 2 ijerph-18-08257-t002:** Simple logistic regression analysis for associated factors related to falls among participants.

Variables	Crude OR ^a^	95% CI ^b^	Wald Statistics	*p*-Value
Age group				
65–69	1.000	-	-	-
70–74	2.027	1.124, 3.656	5.515	0.019
>75	0.932	0.79, 1.811	0.044	0.834
Gender				
Male	1.000	-	-	-
Female	1.033	0.616, 1.731	0.015	0.902
Living arrangement				
With others	1.000	-	-	-
Alone	2.060	0.645, 6.578	1.487	0.223
Medical characteristics				
ECAQ score	1.007	0.799, 1.268	0.003	0.956
Barthel index score	0.970	0.941, 1.001	3.562	0.059
Visual impairment	1.182	0.712, 1.963	0.417	0.519
Polypharmacy	1.766	1.014, 3.077	4.029	0.045
Number of medications used by participants	1.096	0.962, 1.249	1.907	0.167
Number of antihypertensives used by participants	0.779	0.595, 1.020	3.288	0.070
POMA score	0.927	0.877, 0.981	6.968	0.008
Medical comorbidities				
Diabetes mellitus	1.525	0.892, 2.608	2.375	0.123
Heart disease	0.994	0.472, 2.092	0.000	0.986
Stroke	2.047	0.577, 7.263	1.229	0.268
Hyperlipidemia	1.535	0.405, 5.816	0.398	0.528
Arthritis	1.156	0.696, 1.919	0.313	0.576
Types of antihypertensive				
ACE inhibitor	0.966	0.582, 2.356	0.018	0.893
Calcium channel blocker	0.576	0.328, 1.010	3.713	0.054
Diuretics	1.417	0.852, 2.356	1.803	0.179
Angiotensin receptor blocker	0.794	0.409, 1.543	0.461	0.497
Alpha blocker	0.583	0.156, 2.173	0.647	0.421
Beta blocker	0.574	0.305, 1.080	2.967	0.085
Blood pressure (mmHg)				
Sitting				
Systolic blood pressure	0.996	0.989, 1.003	1.267	0.260
Diastolic blood pressure	0.999	0.987, 1.001	0.033	0.865
Standing				
Systolic blood pressure	0.994	0.981, 1.007	0.855	0.355
Diastolic blood pressure	1.008	0.985, 1.003	0.467	0.494
Orthostatic hypotension, n (%)	0.842	0.312, 2.270	0.116	0.734

*p*-value is derived from a one-way ANOVA. ^a^ Crude Odds ratio. ^b^ Confidence interval.

**Table 3 ijerph-18-08257-t003:** Multivariate analysis of the associated factors for falls among hypertensive older adults.

Variable	Adjusted OR ^a^	95% CI ^b^	Wald Statistics	* *p*-Value
Polypharmacy	2.513	1.339, 4.718	8.228	0.004
Diuretics	2.803	1.418, 5.544	8.780	0.003
POMA score	0.940	0.886, 0.996	4.358	0.037
Number of antihypertensives	0.473	0.319, 0.700	14.024	0.000

^a^ Adjusted odds ratio. ^b^ 95% confidence interval. * *p*-value was derived from a multiple logistic regression analysis.

## Data Availability

Data are available on request due to restrictions. The data presented in this study are available on request from the corresponding author. The data are not publicly available due to privacy issues.

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
