# Peer review of "Older Adults with Hypertension: Prevalence of Falls and Their Associated Factors"

_ijerph, 2021, doi:10.3390/ijerph18168257_

Round 1
Reviewer 1 Report
Dear authors
Congratulations for the research “Prevalence of Falls and Their Associated Factors among the
Hypertensive Elderly in East Coast Malaysia”.
The research it is very interesting and the thematic relevant.
Title: Actually, the use of the word “elderly” is no longer recommend. Please adopted the “older adults” “older person” in the title and revised all the manuscript.
Abstract: The authors must check the author guidelines “The abstract should be a single paragraph and should follow the style of structured abstracts, but without headings”. In the conclusion the authors reported that polypharmacy and diuretic were predictors for risk of fall. However, the aim of the study is related to the fall not the risk of fall. In the abstract add information about the population/sample. In the results reported the prevalence of fall (rate of falling), but not the risk of fall.
Introduction: the background could use some more coherent organization. The last paragraph related Hypertension must be after the demographic data. authors should reinforce the relevance of the study, as well as analyze the problem more objectively.
Methods. Provide information about the number of participants.
Why considerer “The elderly is defined as having a chronological age of 65 years or older [23]” and not 60 years or older.
No information about the samples (size, type of sampling). Clarified the inclusion criteria.
“Performance Oriented Mobility Assessment (POMA) is comprised of 9 balance tasks and 82 7 items used to assess their gait characteristics [26]. The balance components are sitting balance, standing, attempts to stand, immediate standing balance, standing balance, nudged, eyes closed, turning 360 degrees, and sitting down. This comes to a total of 16 85 points. The gait elements consisted of a commencement of gait, step length, step symmetry, step continuity, path, trunk and walking attitude for a maximum of 12 points. Each 87 subscale was measured as either abnormal = 0 or normal = 1, or in some cases, adaptive = 88 1 and normal = 2. The total score is correlated with different levels of fall risk from low risk (25–28) to medium or high risk (0–24) [5].” – this information in the section – instrument
Usually, the POMO score ranges from 0 to 28, and the highest risk of falling is present for scores<19.
This is information is not accurate “The total score is correlated with different levels of fall risk from low 89 risk (25–28) to medium or high risk (0–24) [5].”
Provide information about validity and reliability of POMA to Malaysia.
Please clarify this sentence “The Elderly Cognitive Assessment Test is a socio-demographic questionnaire 93 that includes the participant’s socio-demographic data such as age, gender and living arrangement.”
The authors reported seven section in the research instrument, but mention six. More information about the tools used in this section must be provide (e.g. functional status)
The research instruments consisted of a case report that divided the area into seven 92 sections
Data collection with information about instrument. Who performed the data collection? When? Were?
Globally, the methods section needs to be organized, add inform and restructure the different sections.
Please explain the meaning of health clinics. There are different services worldwide. Primary care? Hospital? Inpatients?
Results. Relevant information must be adding to the text before the tables.
Lines 1137 to 143 have the same information. the results in table 2 are poorly described, it is suggested that they report the most relevant statistically significant differences.
More data related to the univariate will be relevant.
The results of ROC was very low (0.327) – this must ne discuss in the limitation section.
Variables such as, diuretics and number of antihypertensive were used to calculation of polypharmacy. How the authors deal with the multicollinearity with polypharmacy variable
Discussion. It is appropriate for the relevance of the discussion to the results and the background of the study. However, the authors reported systematic review and studies with more robust samples that have previously confirmed the results obtained. So, what does this study add to the literature?
Take more Antihypertensives was related to the reduced the risk of fall. However, take more antihypertensive medication contribute to the polypharmacy that is a predictor of higher risk. What the authors thing/analysis about this? The mean of antihypertensive medication? what is the contribution of this medication to polymedication?
The limitation of the study was also be discussed, but need a deep analysis.
Organization and style of presentation: OK
Reference are relevant, but only 35% was published in the last 5 yearsGood luck and thank you very much for giving me a chance to review.
Author Response
Prevalence of Falls and Their Associated Factors among the Hypertensive Older Adults in East Coast Malaysia
Reviewer 1:
Thank you for taking the time to review this manuscript and provide feedback. All of your suggestions for improving our manuscript are highly appreciated.
|
No |
Suggestion/feedback from reviewer |
Authors’ response |
|
1 |
Title: “Prevalence of Falls and Their Associated Factors among the Hypertensive Elderly in East Coast Malaysia”. Actually, the use of the word “elderly” is no longer recommend. Please adopted the “older adults” “older person” in the title and revised all the manuscript. |
We replaced the word ‘elderly’ with ‘older persons’ or ‘older adults’ in our manuscript |
|
2 |
Abstract: The authors must check the author guidelines “The abstract should be a single paragraph and should follow the style of structured abstracts, but without headings”. In the conclusion the authors reported that polypharmacy and diuretic were predictors for risk of fall. However, the aim of the study is related to the fall not the risk of fall. In the abstract add information about the population/sample. In the results reported the prevalence of fall (rate of falling), but not the risk of fall. |
All of the headings in the abstract were removed. Information of population and sample was added in the abstract. Conclusion in the abstract was corrected |
|
3 |
Introduction: the background could use some more coherent organization. The last paragraph related Hypertension must be after the demographic data. authors should reinforce the relevance of the study, as well as analyze the problem more objectively. |
The paragraph regarding hypertension was moved after the demographic data. The relevance of the study was emphasized |
|
4 |
Methods. Provide information about the number of participants.
|
The number of participants was added in the methods section |
|
5 |
Why considerer “The elderly is defined as having a chronological age of 65 years or older [23]” and not 60 years or older.
|
According to a United Nation report, older persons as those aged 60 or 65 years or over.[1] However, the Department of Statistics, Malaysia in its report on Population projection (revision), Malaysia 2010-2040 stated that old age population are those who aged 65 and above [2] |
|
6 |
No information about the samples (size, type of sampling). Clarified the inclusion criteria. |
Sample size and type of sampling were added in the section of study design and population, and data collection. The inclusion criteria - hypertensive older adults who were diagnosed with hypertension for at least 1 year This is because, in our study, we look for a history of falls that occur within the last 1 year -under treatment for at least 2 weeks In the literature, it is stated that most antihypertensive will take effect approximately 2 weeks after their initiation [3]. This inclusion criterion is important because we were also looking for the relation of antihypertensive with falls. |
|
7 |
“Performance Oriented Mobility Assessment (POMA) is comprised of 9 balance tasks and 7 items used to assess their gait characteristics. The balance components are sitting balance, standing, attempts to stand, immediate standing balance, standing balance, nudged, eyes closed, turning 360 degrees, and sitting down. This comes to a total of 16 points. The gait elements consisted of a commencement of gait, step length, step symmetry, step continuity, path, trunk and walking attitude for a maximum of 12 points. Each 87 subscale was measured as either abnormal = 0 or normal = 1, or in some cases, adaptive = 88 1 and normal = 2. The total score is correlated with different levels of fall risk from low risk (25–28) to medium or high risk (0–24) [5].” – this information in the section – instrument Usually, the POMO score ranges from 0 to 28, and the highest risk of falling is present for scores<19. This is information is not accurate “The total score is correlated with different levels of fall risk from low 89 risk (25–28) to medium or high risk (0–24) [5].” Provide information about validity and reliability of POMA to Malaysia.
|
We agree with the reviewer regarding the highest risk of falling is present for scores<19. However, in this study, we analysed the mean of the POMA score. This statement ‘The total score is correlated with different levels of fall risk from low 89 risk (25–28) to medium or high risk (0–24)’ was omitted from the manuscript. Based on our reading and searching, there is no available information regarding the validity and reliability of POMA to Malaysia. However, there was a previous study of falls among Malaysian population that using POMA as their assessment tool [4]. |
|
8 |
Please clarify this sentence “The Elderly Cognitive Assessment Test is a socio-demographic questionnaire 93 that includes the participant’s socio-demographic data such as age, gender and living arrangement |
We already corrected this statement and clarify further regarding our case report form in the research instrument section. |
|
9 |
The authors reported seven section in the research instrument, but mention six. More information about the tools used in this section must be provide (e.g. functional status)
|
We truly apologize for the wrong statement regarding the number of sections in the case report form and we already corrected the statement. More information regarding tools are added in the method section
|
|
10 |
Data collection with information about instrument. Who performed the data collection? When? Were? |
Further detailed collection information is added |
|
11 |
Globally, the methods section needs to be organized, add inform and restructure the different sections.
|
The method section had been restructured into study design and population, data collection and statistical analysis. |
|
12 |
Please explain the meaning of health clinics. There are different services worldwide. Primary care? Hospital? Inpatients?
|
The term ‘health clinic’ in this manuscript means that primary health clinics. All of the ‘health clinic’ terms had been changes to ‘primary health clinic’. For health system in Malaysia, it is divided into 3 classifications: Primary: public primary health clinic Secondary: smaller public hospitals Tertiary: more complex tertiary care, in regional and national hospitals [5] |
|
13 |
Results. Relevant information must be adding to the text before the tables. Lines 1There is a more in-depth explanation of those tables included.There is a more in-depth explanation of those tables included.37 to 143 have the same information. the results in table 2 are poorly described, it is suggested that they report the most relevant statistically significant differences. More data related to the univariate will be relevant.
|
Table 1 and table 2 had been improved to add more information regarding the variables. A more detailed explanation regarding that table is included. |
|
14 |
The results of ROC was very low (0.327) – this must be discuss in the limitation section. |
The low ROC (0.327) had been discussed in the limitation section |
|
15 |
Variables such as, diuretics and number of antihypertensive were used to calculation of polypharmacy. How the authors deal with the multicollinearity with polypharmacy variable |
We were aware of the multicollinearity when diuretics and the number of antihypertensive were used for the calculation of polypharmacy. However, for clinical practice purposes, we conducted data analysis on antihypertensive types and numbers of antihypertensive. Interestingly, our findings revealed that the number of antihypertensives taken is unrelated to fall incidence, which opposes polypharmacy practice. |
|
16 |
Discussion: The authors reported systematic review and studies with more robust samples that have previously confirmed the results obtained. So, what does this study add to the literature?
|
This study adds new information for Malaysian older person data. -Prevalence of falls among hypertensive older persons in Malaysia - Orthostatic hypotension incidence was low among this group - The number of antihypertensives was reported to be protective towards falls Line 339-349 Further explanation regarding these findings was elaborated in the discussion |
|
17 |
Take more Antihypertensives was related to the reduced the risk of fall. However, take more antihypertensive medication contribute to the polypharmacy that is a predictor of higher risk. What the authors thing/analysis about this? The mean of antihypertensive medication? what is the contribution of this medication to polymedication? |
We agreed that taking more antihypertensive contributes to polypharmacy practices, which is also a predictor of falls. In this matter, we tried to look into the use of antihypertension. Maybe further research is needed to analyse the types of drugs that contribute to falls among those older persons with polypharmacy instead of just looking into the numbers of medications.
This is new to our knowledge that a higher number of antihypertensive is protective towards fall, thus this information can be used as a guide for a medical practitioner in managing hypertension among older adults |
|
18 |
The limitation of the study was also be discussed, but need a deep analysis. |
We had made some improvements to the limitation section |
|
19 |
Reference are relevant, but only 35% was published in the last 5 years |
We had changed some of the references to more recent references. The percentage of references that were published within the last 5 years is 53%. |
References:
- United Nation, D.o.E.a.S.A., Population Division. World Population Ageing 2019 ST/ESA/SER.A/444 2020; Available from: https://www.un.org/en/development/desa/population/publications/pdf/ageing/WorldPopulationAgeing2019-Report.pdf.
- Mahidin, M.U., Current Population Estimates, Malaysia, 2020. 2020, Department of Statistics Malaysia.
- Butt, D., et al., The risk of falls on initiation of antihypertensive drugs in the elderly. Osteoporosis international, 2013. 24(10): p. 2649-2657.
- Azidah, A., H. Hasniza, and E. Zunaina, Prevalence of falls and its associated factors among elderly diabetes in a tertiary center, Malaysia. Current gerontology and geriatrics research, 2012. 2012.
- Organization, W.H., Malaysia health system review. 2012, Manila: WHO Regional Office for the Western Pacific.

Reviewer 2 Report
This study examined the fall rate and related factors among hypertension elderly in Malaysia. Here are some comments or suggestions for the authors.
- To compare the falling prevalence of older people with and without hypertension, I suggest the average falling rate among the general older people should be cited in the literature. And then it is possible to conclude whether the falling rate of hypertension in older adults (32.2%) was high or low in Malaysia. Then the significance of this paper can be shown.
- Polypharmacy and diuretic usage may be related to falls. Because multi-morbidity is often found in the older population, whether the multiple medicine use is appropriate needs to be examined carefully. Maybe some of the medicines are not necessary or have interactions with the patients. I wonder if it is possible to examine the antihypertensives combined with other medicines used by the participants can be identified, and possibly to provide some hints about the appropriateness of medicine use in the risk of falls.
- Regarding implications, some of the risk factors of falls were unavailable, such as the environmental risks or the comorbidity not included in this study. This point can be included in the limitation.
- Implications or suggestions based on the results should be provided in the Conclusion.
Author Response
Prevalence of Falls and Their Associated Factors among the Hypertensive Older Adults in East Coast Malaysia
Reviewer 2:
Thank you for taking the time to review this manuscript and provide feedback. All of your suggestions for improving our manuscript are greatly appreciated.
|
No |
Suggestion/feedback from reviewer |
Author’s response |
|
1 |
To compare the falling prevalence of older people with and without hypertension, I suggest the average falling rate among the general older people should be cited in the literature. And then it is possible to conclude whether the falling rate of hypertension in older adults (32.2%) was high or low in Malaysia. Then the significance of this paper can be shown.
|
The prevalence of falls among the general population of older adults was provided in the introduction section. We have concluded that the prevalence of falls among hypertensive older adults in Malaysia is significantly high. |
|
2 |
Polypharmacy and diuretic usage may be related to falls. Because multi-morbidity is often found in the older population, whether the multiple medicine use is appropriate needs to be examined carefully. Maybe some of the medicines are not necessary or have interactions with the patients. I wonder if it is possible to examine the antihypertensives combined with other medicines used by the participants can be identified, and possibly to provide some hints about the appropriateness of medicine use in the risk of falls.
|
It is undeniable that older adults are usually having multiple comorbidities. However, in this study, we are unable to provide information regarding the necessity or interaction of the medication that was taken by our participants. This is because the information was not taken during the research data collection. |
|
3 |
Regarding implications, some of the risk factors of falls were unavailable, such as the environmental risks or the comorbidity not included in this study. This point can be included in the limitation. |
We agree that some of the falls predictors were not investigated in this study, thus we had included this in the limitation section |
|
4 |
Implications or suggestions based on the results should be provided in the Conclusion. |
We had provided the implication and further suggestions in conclusion based on the result. |

Round 2
Reviewer 1 Report
Dear Authors
Thank you for the revision and attend all the comments and questions.
I suggest a final edition of Engish language